# Effect of the Zwitterion, p(MAO-DMPA), on the Internal Structure, Fouling Characteristics, and Dye Rejection Mechanism of PVDF Membranes

**DOI:** 10.3390/membranes10110323

**Published:** 2020-10-31

**Authors:** Nelisa Ncumisa Gaxela, Philiswa Nosizo Nomngongo, Richard Motlhaletsi Moutloali

**Affiliations:** 1Department of Chemical Sciences, University of Johannesburg, Doornfontein Campus, P.O. Box 17011, Johannesburg 2028, South Africa; nelisa.gaxela@gmail.com (N.N.G.); pnnomngongo@uj.ac.za (P.N.N.); 2DSI/Mintek Nanotechnology Innovation Centre, Water Research Node P.O. Box 17011, Doornfontein, Johannesburg 2028, South Africa; 3DSI/NRF SARChI: Nanotechnology for Water, University of Johannesburg, Doornfontein 2028, South Africa

**Keywords:** zwitterions, polyvinylidene fluoride, p(MAO-DMPA), antifouling, dye rejection

## Abstract

The zwitterion poly-(maleic anhydride-alt-1-octadecene-3-(dimethylamino)-1-propylamine) (p(MAO-DMPA)) synthesized using a ring-opening reaction was used as a poly(vinylidene fluoride) (PVDF) membrane modifier/additive during phase inversion process. The zwitterion was characterized using proton nuclear magnetic resonance (^1^HNMR) and attenuated total reflectance Fourier transform infrared spectroscopy (ATR-FTIR). Atomic force microscopy (AFM), field emission scanning electron microscope (SEM), FTIR, and contact angle measurements were taken for the membranes. The effect of the zwitterionization content on membrane performance indicators such as pure water flux, membrane fouling, and dye rejection was investigated. The morphology of the membranes showed that the increase in the zwitterion amount led to a general decrease in pore size with a concomitant increase in the number of membrane surface pores. The surface roughness was not particularly affected by the amount of the additive; however, the internal structure was greatly influenced, leading to varying rejection mechanisms for the larger dye molecule. On the other hand, the wettability of the membranes initially decreased with increasing content to a certain point and then increased as the membrane homogeneity changed at higher zwitterion percentages. Flux and fouling properties were enhanced through the addition of zwitterion compared to the pristine PVDF membrane. The high (>90%) rejection of anionic dye, Congo red, indicated that these membranes behaved as ultrafiltration (UF). In comparison, the cationic dye, rhodamine 6G, was only rejected to <70%, with rejection being predominantly electrostatic-based. This work shows that zwitterion addition imparted good membrane performance to PVDF membranes up to an optimum content whereby membrane homogeneity was compromised, leading to poor performance at its higher loading.

## 1. Introduction

The textile industry is one of the largest and most important industries in the world [1]. However, the wastewater from this industry poses serious challenges to the environment and is considered a source of harmful pollutants due to the diversity of dyes [2]. In addition to the textile industry, paper, plastic, pharmaceuticals, cosmetics, and beverages also contribute to discharging wastewater containing different kinds of dyes [3]. When these dyes are released to the environment, they pose a great threat to ecosystems and to public health [4,5]. It has been proven that dyes are toxic and negatively affect the aesthetic of water and also reduce the photosynthetic activity of aquatic organisms even at low concentrations [6]. The lakes and rivers that these pollutants flow through are prone to eutrophication in the long run [7]. In trying to remove these toxic pollutants, numerous conventional methods have been studied. These include adsorption [8,9], coagulation [10,11], biological degradation [12,13], and advanced oxidation processes [14,15]. These methods display drawbacks such as insufficient dye removal, unsustainability with regard to dye recovery, and production of secondary pollution [16]. For instance, the advantages of the widely used adsorption technology are simplicity, flexibility, and the use of different types of adsorbents, with its main and critical drawback being the production of secondary pollution in the form of sludge loaded with toxic dyes [17]. This leads to more expensive post-treatment processes, which is a major challenge when the adsorption process is used at a larger scale [18]. To address the challenges posed by the above-mentioned technologies, a microbial fuel cell to directly convert dyes from wastewater into bioelectricity while desalinating the wastewater has been explored [19]. However, it was determined that this technique was also not feasible, as it rendered it hard for the valuable residuals of these dyes to be recovered [20]. Therefore, there is an urgent need to develop cost-effective technologies for the treatment of wastewater containing dyes. 

The use of membrane technology has become widely applied in various fields such as water treatment [21], gas purification [22], food processing [23], pharmaceuticals [24], and environmental protection [25]. Membranes play an important role in wastewater treatment. They are utilized for oily wastewater separation [26], biological and chemical oxygen demand (COD and BOD) reduction [27], heavy metal ion removal [28], textile wastewater treatment [29], and nuclear waste treatment [30]. Over the years, literature has highlighted properties that make good membranes, such as higher flexibility, smaller footprint demand for installation, better control of the pore-forming mechanism, and low costs [31]. Membranes for industrial processes have been made from both inorganic and organic polymers, with the organic-based being the dominant ones. These organic polymers include polysulfone (PSF) [32,33,34,35], polyacrylonitrile (PAN) [36,37,38,39], polyethersulfone (PES) [40,41,42,43], and poly(vinylidene fluoride) (PVDF) [44,45,46,47], among others. These membranes each have distinct characteristics that make them good candidates for the desired improvements and applications [24,48]. 

Membranes made from poly(vinylidene fluoride) PVDF, a semi-crystalline polymer with repeated units of –(CH_2_CF_2_)*_n_*–, are also produced via conventional non-solvent induced phase separation [24]. In addition to its chemical and thermal stability and high mechanical strength [27], PVDF is one of the most widely used polymers in membrane technology for separation applications [23,49]. Even with so many good qualities, the hydrophobic nature of the PVDF membrane makes it prone to fouling, which is caused by the deposition and accumulation of constituents in the feed stream of the membrane [50]. This is a long-term critical issue that occurs particularly in the PVDF membrane-related applications, thereby decreasing its performance and efficiency [24,27]. It has been reported that an increase in hydrophilicity offers membrane fouling resistance [51,52]. In trying to rectify this issue, Xu et. al. used graphene oxide and silane functionalized graphene to enhance the hydrophilicity of PVDF membranes for better antifouling resistance [53]. In another study, Maziya et al. grafted silver nanoparticles onto hyperbranched polymer nanofibrous membranes in order to enhance its dual antifouling properties [54]. Other researchers have reported the incorporation of different nanomaterials to increase membrane hydrophilicity [55,56,57]. 

Previously, poly(ethylene glycol) (PEG) materials and its derivatives have been widely used to enhance hydrophilic properties and reduce fouling [58,59]. This is because poly(ethylene glycol) (PEG) materials possess hydration, are easily controlled, and have biocompatibility [60]. The downside is that due to oxidation, PEG tends to be very unstable in the presence of oxygen and transition metals [40,45]. More recently, studies have proven that zwitterionic molecules are effective antifouling materials due to their hydrophilicity rendering excellent antifouling ability to membranes [61,62,63,64]. Zwitterionic polymers are polymers that simultaneously contain both the negatively and positively charged ions, which are arranged as the pendent-side chain structure [58]. This unique molecular structure provides them with excellent properties that allow them to bind water molecules more strongly than PEG materials [49,65]. The excellent antifouling properties imparted through the incorporation of zwitterionic molecules have led to their exploration as the most effective antifouling medium and have been shown to be ultra-low fouling even when in contact with complex media such as an undiluted human plasma or serum [66]. These materials can be incorporated into membranes via coating [23,49], grafting [44,61], or blending [67,68]. Coating and grafting are two-step processes that take place at the surface of the membrane. Blending, however, is a one-step process of membrane modification that is done during membrane preparation; hence, lower membrane production costs are realized than through the other processes [51]. Careful selection of zwitterion modifiers is important to minimize their leaching during applications; for instance, zwitterions with sidechains are known to interact strongly with PVDF polymers, thereby minimizing their leaching [69]. Thus, the zwitterion selected in this study is expected not to leach in line with this expectation. The introduction of hydrophilic modifiers is said to enhance the performance, thereby removing pollutants while prolonging the life of the membrane.

The effects of the incorporation of zwitterion, p(MAO-DMPA), into a PVDF polymer matrix during phase inversion are explored and reported herein. It is envisaged that the PVDF ultrafiltration membrane produced in this manner will possess enhanced physicochemical properties, namely porosity, hydrophilicity, and water uptake capacity, leading to improved performance indicators such as high flux, solute rejection through electrostatic interaction, and antifouling properties. The preparation of p(MAO-DMPA) and its incorporation into a PVDF matrix via phase inversion has not been previously reported and its effects on the modified membranes on water flux and solute rejection has not yet been studied for wastewater applications. Moreover, their functionalities will impart additional mechanisms of solute interaction that will lead to additional solute mechanism rejection.

## 2. Materials and Methods

### 2.1. Materials

Poly(vinylidene fluoride) (PVDF), maleic anhydride-alt-1-octadecene (MAO), 3-(dimethylamino)-1-propylamine (DMPA), tetrahydrofuran (THF), *N*-methyl-2-pyrrolidone (NMP) anhydrous, poly(ethylene glycol) (PEG), Congo red (CR), Rhodamine 6G (RG), and bovine serum albumin (BSA) were purchased from Sigma-Aldrich (Johannesburg, South Africa). All materials were used as purchased with no further purification.

### 2.2. Synthesis of p(MAO-DMPA)

The zwitterionic polymers were synthesized according to the method of Venault et al. (Figure 1) [23]. In short, MAO (0.01 mol) and DMPA (0.32 mol) were stirred in THF (50 mL) for 20 min at room temperature (RT). The mixture was then centrifuged three times at 5000 revolutions per minute (rpm) for 30 min to separate the resultant polymer from the solvent. The polymer was then dried under vacuum and stored at 4 °C until use. The formation of zwitterion was confirmed by ^1^H-NMR and FTIR. 

### 2.3. Preparation of Membranes

The PVDF membranes were synthesized following the procedure reported by Ahmad et al. [70] with slight modifications. A series of PVDF membranes with different compositions (wt.%) of p(MAO-DMPA) and NMP were prepared (Table 1). The solvent, NMP (80–82 mL), the polymer (16 g), and the appropriate amount of p(MAO-DMPA) (0.5–2.0 g) were stirred in a beaker using an overhead stirrer until a homogeneous casting solution was obtained, at about 24 h. The casting solution was then placed under vacuum for 24 h to remove dissolved gases and obvious air bubbles. The membranes were formed by casting the solution onto a glass plate with a blade set at a 200-µm air gap. The polymer film was then immersed in a coagulation bath at room temperature and left for 24 h. The formed membranes were transferred into fresh distilled water and stored in the fridge until needed.

### 2.4. Characterization of the Membranes 

The surface and cross-section morphology of membranes was analyzed using a field emission scanning electron microscope (SEM, VEGA 3 TESCAN, a.s., Brno, Czesh Republic) at the acceleration voltage of 20 kV. To obtain the membrane cross-section, the membranes were immersed in liquid nitrogen and fractured whilst hard. All the samples were coated with carbon before observation to reduce charging. Attenuated total reflectance Fourier transform infrared spectroscopy (ATR-FTIR, Perkin Elmer Spectrum 100 Spectrometer, Bruker, Karlsruhe, Germany) was used to determine the surface composition of PVDF membranes between 650–4000 cm^−1^ with 4 cm^−1^ resolution. Water contact angle (WCA) was measured using DataPhysics Optical Contact Angle (COCA) 15 EC (KRUSS, Hamburg, Germany) equipped with video capture at room temperature to evaluate the surface wetting ability using the sessile drop shape image analysis system. A dosing volume of 1 µL and a fast dosing rate with no continuous dosing were utilized. The readings were taken from at least five spots on the same membrane, and the values were then averaged out. The quantitative surface roughness analysis of the PVDF membranes was measured using atomic force microscopy (AFM, Nanoscale IV, Veeco, California, USA) with the spring constant of 0.12 N·m^−1^ through the contact mode in dry air. All the membranes were dried for 6 h in a vacuum oven before the AFM analysis was performed. The concentration of the dyes in the permeate was determined using UV-vis spectrophotometer (UV-2450, Shimadzu, Suzhou Jiangsu, China) in the wavelength range between 190 and 800 nm. 

Water uptake and porosity studies were performed on all membranes following reported methods [71]. In short, the amount of water taken in by the membrane was determined by cutting the membrane and soaking it in water for 24 h. The weight was recorded as *M_w_* and then dried to record the *M_d_*. The calculation of the water uptake was then determined using Equation (1):(1)$$Water\;uptake\;rate\;\left(\%\right)=\frac{\left(M_{w}-M_{d}\right)}{M_{d}}\;\times \;100$$

The porosity of the membrane was determined by gravimetric analysis using Equation (2):(2)$$porosity,\;\varepsilon =\frac{M_{w}-M_{d}}{\rho_{w}\cdot A\cdot L}$$
where *M_w_* and *M_d_* represent the wet and dry membrane weights (g), *ρ_w_* is the density of water (g·m^−3^) at RT, *A* is the surface area of the membrane (m^−2^), and *L* is the thickness of the membrane (m).

### 2.5. Membrane Performance Assessment

A dead-end stirred cell filtration system connected to a nitrogen gas cylinder was utilized to study the filtration performance of the membranes. All membranes were initially compacted for 30 min with deionized water at 300 kPa. Thereafter, the membrane performance was assessed at various applied pressures to obtain flux, rejection, and fouling parameters. The water flux (*J*) and rejection (*R*) were calculated using Equations (3) and (4), respectively: (3)$$J=\frac{V}{A\;\times \;t}$$
where *V* is the volume (L) of the permeated water, *A* is the membrane area (m^2^), and *t* is the permeation time (h);
(4)$$R\;\left(\%\right)=\left(1-\frac{C_{p}}{C_{f}}\right)\;\times \;100$$
where *C_p_* and *C_f_* (mg·mL^−1^) represent the concentration of the permeate and the feed solution, respectively. 

Fouling resistance of the membranes was assessed by measuring pure water flux (*J_w_*_1_) followed by that obtained during filtration of a BSA (1000 mg·mL^−1^) feed solution (*J_p_*). The membranes were subsequently rinsed with deionized water for 1 h using the backwashing method. It is noted here that the membranes did not show any signs of damage and maintained their integrity after this process. Pure water flux (*J_w_*_2_) was again measured through the rinsed membrane to determine the flux recovery. The flux recovery ratio (*FRR*), total fouling ratio (*R_t_*), reversible fouling ratio (*R_r_*), and irreversible fouling ratio (*R_ir_*) of the membranes were calculated using the Equations (5)–(8), respectively.
(5)$$FRR\;\left(\%\right)=\left(\frac{J_{w2}}{J_{w1}}\right)\times 100$$
(6)$$R_{t}\;\left(\%\right)=\left(1-\frac{J_{p}}{J_{w1}}\right)\times 100$$
(7)$$R_{r}\;\left(\%\right)=\left(\frac{J_{w2\;}-J_{p}}{J_{w1}}\right)\times 100$$
(8)$$R_{ir}\;\left(\%\right)=\left(\frac{J_{w1\;}-\;J_{w2\;}}{J_{w1}}\right)\times 100\;=\;R_{t}\;-\;R_{r}$$
where *J_w_*_1_ is the pure water flux before the fouling run, *J_w_*_2_ is the water flux after washing the fouled membrane, and *J_p_* is the flux during the BSA filtration run [5].

## 3. Results and Discussion

### 3.1. Characterization of the Zwitterion, p(MAO-DMPA)

The zwitterion p(MAO-DMPA) was synthesized using a ring-opening polymerization of the poly(maleic anhydride-alt-1-octadecene) while reacting with the 3-dimethylamino-1-propylamine in the presence of THF (Figure 1). The formation of the zwitterion was achieved through the condensation reaction between the propylamine and the anhydride ring’s functional group. The successful formation was confirmed with the aid of ^1^H-NMR and FTIR spectroscopy.

The chemical structure of zwitterion p(MAO-DMPA) [23,51] was confirmed by ^1^HNMR spectroscopy (Figure 2), which was in agreement with literature reports. For instance, the NMR spectrum of the zwitterion p(MAO-DMPA) showed two characteristic peaks at 3.7 ppm and 1.2 ppm assigned to the methyl group linked to the N-atom of the quaternary amine and terminal methyl group, respectively. 

On the other hand, in the emergence of the amide band around 1634 cm^−1^ [51] with a concomitant broadening of the amine band confirmed the formation of the target zwitterion. Peaks observed at 3293 cm^−1^ and 3363 cm^−1^ are indicative of the presence of the stretching vibrations of the amine group in the DMPA fragment (Figure 3). 

### 3.2. Characterization and Membrane Performance of the Ultrafiltration Membranes

#### 3.2.1. FTIR Analysis

The FTIR spectra showing the functional groups of the pristine PVDF and PVDF/p(MAO-DMPA) blend membranes are shown in Figure 4. The presence and relative abundance of the zwitterion in the membrane formulation is confirmed through the increase in the band at ca. 2857 cm^−1^ attributed to the CH_2_ of the alkane chain. In addition, the characteristic band attributed to the amide group at 1671 cm^−1^ also confirms the presence of the zwitterion. The slight shift in zwitterion functional groups compared to those of the free zwitterion (i.e., 2852 cm^−1^, 2920 cm^−1^, and 1634 cm^−1^) is indicative of some interaction between the zwitterion and the PVDF polymer chains in the membrane matrix. The PVDF polymer bands are as expected; for instance, the bands around 840 cm^−1^ in all spectra were attributed to plane bending or rocking vibration in α phase of PVDF polymer, whilst the bands around 872 cm^−1^ were attributed as the mixed mode of CH_2_ rocking and CF_2_ asymmetric stretching in β and γ phase of the PVDF polymer [72]. Bands around 1170 cm^−1^ were attributed to the asymmetrical stretching of the CF_2_ group, while the bands at 1272 cm^−1^ are attributed to the γ phase [49,72]. These results confirm the successful blending of the zwitterion into the membrane matrix [44,45,51]. 

#### 3.2.2. SEM Analysis

The surface and cross-section morphology as well as the porosity of the membranes were examined using SEM. The surface had a typically porous structure expected for membranes prepared through phase inversion, with little obvious variations with increasing zwitterion content. The surface porosity seemed to increase slightly with increasing zwitterion content from 2.08 to 4.21 as well in line with the decreasing top layer (Figure 5). On the other hand, the water uptake values increased with increasing zwitterion dosages but decreased drastically for MZ3 (1.5%). The water uptake capacity of 320% and 982% was observed for virgin PVDF and the 1.0% p(MAO-DMPA)/PVDF membrane, respectively (Figure 6). These values complement the results observed for contact angle (Figure 7), deducing that hydrophilicity enhances the rate of water uptake [70], as well as SEM observations relating to the relative membrane internal structure (Figure 5a’–e’) (i.e., mutation from sponge-like structures to interlinked networks) [72]. The uptake ratio tends to increase with increasing zwitterion incorporation, but in the current scenario, the higher dosages resulted in increased surface heterogeneity that negatively affected the membrane’s ability to trap more water molecules (Table 2) [51]. The cross-section of pristine PVDF (Figure 5a’) exhibits a sponge-like morphology with a wide finger-like structure in line with previous reports [45]. A subsequent increase in zwitterion content in the PVDF matrix caused a noticeable change in both the sponge- and the finger-like morphology. For instance, the finger-like structures morphed into an interlinked network of channels with increasing zwitterion addition (Figure 5b’–e’), with the top layer becoming even thinner. The disappearance of the sponge-like morphology leaving only finger-like or, in this case, interlinked channels is attributed to an instantaneous demixing induced by the presence of the zwitterion in the casting solution [52]. The increase in (MAO-DMPA) content resulted in relatively porous membranes compared to the pristine PVDF membrane, which is in line with previous observations [72]. The change in morphology at higher zwitterion content is most likely due to increased immiscibility of the additive in the PVDF matrix, a phenomenon that was previously observed at higher loadings [73].

#### 3.2.3. AFM Analysis

The variation of the membrane surface roughness with varying amounts of incorporated p(MAO-DMPA) is shown in Figure 8, where the nodules or the brightest points are elevated areas, whilst the darker areas represent depressions on the membrane surfaces indicative of surface pores [71]. The surface roughness is reflected in the relative values of R_a_ and R_q_ parameters, the arithmetic mean deviation of roughness that is obtained by measuring the peak heights from one-dimensional plane and the root mean square Z-data, respectively [74]. It was observed that there was a general increase in these R parameters as the zwitterion content was increased. There was, however, a decrease in these parameters for the membrane with the highest zwitterion content (Table 3). This observed decrease is consistent with the trends observed earlier for the membrane structure and porosity in the SEM analysis (Figure 5). The overall trend is that increasing the zwitterion content seems to lead to denser membranes with smoother surfaces and uniformly dispersed surface pores. The smoother surface is expected to be a key contributing factor in enhancing the anti-fouling properties of the prepared membranes [35]. In fact, it was previously reported that membranes with loose internal structures tended to be more prone to fouling than those of a denser nature [75]. It is therefore envisaged that the membrane fouling profile will be improved with increasing zwitterion content. 

#### 3.2.4. Water Contact Angle Analysis

The surface hydrophilicity of the membranes was assessed using a sessile drop contact angle (CA) measurement technique. The variation of the CA is captured in Figure 7, revealing that all the membranes were moderately hydrophilic with CAs below 90° but higher than 60° [5,71]. The CA decreased with the incorporation of the hydrophilic zwitterions, with the 1 wt.% blended membrane having the lowest angle of 63° whereafter it started to rise again, with the 1.5 and 2.0 wt.% membrane having the highest contact angles. The reversal in CA trend at higher zwitterion content can be attributed to the increase in surface/membrane heterogeneity at higher zwitterion content, a phenomenon that was previously reported [51]. A similar reversal at higher zwitterion content was also reported by Dong et al. [76]. In addition, prior reports also showed that the incorporation of zwitterion into the PVDF matrix does not affect the CA greatly [23]. For instance, blending the zwitterion into the PVDF casting solution resulted in CAs that were still relatively higher when compared with other base polymers such as PA [43], PSF [77], PAN [37], and PES [40].

#### 3.2.5. Water Uptake and Porosity Studies

The water uptake values (Equation (1)) increased with the increasing zwitterion dosages but decreased drastically for MZ3 (1.5%). The water uptake capacity of the membranes ranged between 320% and 982% for the virgin PVDF and the 1.0% p(MAO-DMPA/PVDF membrane, respectively (Figure 6). These values complement the results observed for contact angle (Figure 7) in that the most hydrophilic membrane exhibited the highest water uptake [70]. In addition, the internal membrane structure also seemed to play a role in the water uptake trends (Figure 5a’–e’) (i.e., mutation from sponge-like structures to interlinked networks) [72]. The uptake ratio tends to increase with increasing zwitterion incorporation, but in our case, the higher dosages resulted in surface heterogeneity, thereby affecting the membrane’s ability to effectively trap more water molecules [51]. The estimated membrane porosity varied between 2.08% and 4.21% with pore sizes ranging between 0.150 to 0.342 μm, indicating that these were microfiltration membranes. Thus, they are expected to have a low dye rejection profile that will allow the effect of the zwitterions to be clearly observed.

### 3.3. Membrane Permeation Flux

The membrane performance indicators (i.e., pure water flux, fouling propensity as indicated using flux recovery, and solute rejection) were assessed using a dead-end filtration cell. The pure water flux increased with the increasing zwitterion content (Figure 9) in agreement with previous reports [35,78]. This is also in line with our expectations, since both membrane hydrophilicity and membrane porosity were positively influenced by increasing the zwitterion content [45]. The permeability of the membranes increased steadily with increasing zwitterion content ((M0) 2.7174 < 3.0884 < 5.2381 < 5.1927 < 8.4774 (MZ4)), showing the influence the zwitterion had on the overall water passage [73].

### 3.4. Dye Rejection Studies

Dyes are found in several industries that deal with coloring, such as the paper, plastic, food, cosmetics, and clothing industries. These dyes have a negative impact on the environment and need to be removed or recycled from industrial wastewater before being discharged into the environment [3]. The effectiveness of the membranes for dye rejection was assessed using two model dyes: Congo red (CR, an anionic dye (C_32_H_22_N_6_Na_2_O_6_S_2_, M_w_ = 696.665 g·mol^−1^)) and rhodamine 6G (RG, a cationic dye C_28_H_31_N_2_O_3_Cl, M_w_ = 479.01 g·mol^−1^)). In the present study, CR dye was used to establish or confirm that the assessed membranes behaved as ultrafiltration membranes [5]. This apparent ultrafiltration (UF) behavior was observed even though the measured pore sizes were in the microporous membrane range of 0.1 to 0.3 μm (Table 2). On the other hand, RG was selected as it is the most widely used dye compound that is very toxic [79]. The opposite charges and different molecular weight also help to determine the selectivity of the prepared membranes. Figure 10a shows that the pristine PVDF membrane had approximately 85% CR dye rejection, indicating that it is a microfiltration (MF) membrane. Figure 10 further shows that of the composite membranes, only MZ1 (0.5 wt.%, ε = 2.71%) and MZ4 (2.0 wt%, ε = 2.30%) exhibited UF membrane characteristics with respect to CR rejection (i.e., above 90%) [5]. The intermediate zwitterion loaded membranes showed lower CR rejections of 38% and 55% for MZ2 (1.0 wt.%, ε = 3.10%) and MZ3 (1.5 wt.%, ε = 4.21%), respectively, which is typical of MF membranes. It is postulated that this is related to both the porosity (ε) and internal membrane structure. For instance, MZ1 has the tightest skin layer, whilst MZ4 has the highest tortuosity, whereas MZ2 and MZ3 are in between. Thus, the CR rejection occurs through two different mechanisms in these membranes, both related to size exclusion, with the effect of solute interaction with zwitterions becoming apparent as the content of the zwitterion increased. On the other hand, the smaller cationic dye, rhodamine 6G, showed a different rejection profile with varying zwitterion content. Whereas all membranes showed lower rejection for the smaller dye (all below 60%), the observed dye rejection increased with an increasing amount of zwitterion content (Figure 10). This indicates that the rejection is not size related, as observed for the larger CR molecule, alluding to a different rejection mechanism. It is postulated that the long alkane chain is closely associated with the PVDF molecule, leaving the positive ends sticking out, whilst the negative sites are close to the polymer backbone, leading to electrostatic repulsive interaction with the positive dye [4,6]. Thus, the relative increase in repulsive interaction as the zwitterion content was increased resulted in increasing rhodamine 6G rejection. It is worth noting that this mechanism is observed or prominent at moderate dye rejections. These two proposed mechanisms therefore offer an explanation for the observed rejection profiles for the two probe dye molecules [52]. 

Increasing the applied pressure (100, 150, and 200 KPa) led to an increase in dye permeation through the membrane, as expected. The best zwitterion membrane (MZ1 and MZ4) still had CR dye rejection above 80% at the highest applied pressure used. An interesting observation was the quantum of the decrease in rejection for the two membranes; for MZ1, this was minimal, whilst for MZ4, it was higher (at about 10%), further confirming the different rejection mechanisms alluded to earlier. The RG dye rejection was decreased to between 45% and 25% for all other membranes from a high of 59% at 100 kPa. For the best membrane, MZ4, the RG rejection decreased from 60% to 45% for 100 to 300 kPa applied pressure, which was the largest drop amongst all the tested membranes. The increased dye permeation with increasing pressure is due to the enhanced concentration polarization effect and minimized general interaction with the membrane surface [20]. The observed solute rejection profile for both the larger dye and the smaller dye with increasing zwitterion content point to the importance of the interaction potential with the dye molecules in offering an additional rejection mechanism for these membranes, as their performance, in some instances, approach that of UF membranes.

### 3.5. Antifouling Performance

Fouling occurs when the pores of the membrane become blocked due to the adsorption or deposition of dissolved solutes that were being filtered, leading to the reduction of the filtration performance of the membrane [80]. Fouling inevitably occurs with continuous membrane filtration through different mechanisms, a process that can be reversible or irreversible. Reversible fouling is when a foulant is partially binding to the membrane and hence can be removed by backwashing. Irreversible fouling, however, is when a foulant is fully bound to the membrane surface and requires chemical cleaning [5]. Due to the cleaning methods necessary to reverse the effects of these mechanisms, reversible fouling is easier to deal with in filtration applications. The membranes showed no signs of damage, as their integrity was intact even after this prolonged backwash process. This resilience offer promise for practical applications. Figure 11a shows that a drastic decrease in the flux was observed when pure water was replaced with a BSA solution as the feed as a consequence of protein fouling or deposition on the membrane surface. It also shows that after simple washing of the membrane with pure water, the water flux was recovered. The extent of the recovery is indicative of the fouling propensity of the membrane under assessment. The corresponding flux recovery ratio (*FRR*), Figure 11b) increased as the zwitterion content increased and seems to track the flux trend (Figure 9). In contrast, the *FRR* values seem to be inversely related to the membrane surface pore sizes (i.e., the higher the pore sizes, the higher the fouling propensity) [81]. The recovery of the pure water flux was highest for MZ3 (0.150 pore size: >95%) and worst for MZ1 (0.235 pore size: ~22%). The cake layer only formed on the surface and did not go deep into the membrane due to the smaller pore sizes observed in Figure 5 but was able to penetrate the larger pores wherein it was difficult to dislodge. Thus, membranes with intermediate zwitterion content that had reduced surface roughness (Figure 8) seem to be far better than those with the least or the most zwitterion content with relatively higher surface roughness [82]. The influence of surface roughness is reported to be important with hydrophobic surface characteristics, which is similar to the current observations for MZ4 compared to others [83,84]. It is also evident that membranes with higher solute rejection fared worse than those with lesser rejection when their FRR values were compared (Figure 11b).

The flux recovery ratio was best for the membranes with intermediate rejection abilities, with those with 1.0 and 1.5 wt.% zwitterion content exhibiting almost total recovery (MZ3) (Figure 10b). Figure 10 further confirms that the intermediate zwitterion content membranes (1.0 and 1.5 wt.%) exhibited higher reversible fouling (*R_r_*) compared to irreversible fouling (*R_ir_*), which is indicative of their greater fouling resistance [85]. A larger proportion of R*_ir_* is indicative of strong entrapment of foulants on the membrane surface and is generally prevalent in membranes with larger surface pores [81]. In terms of this indicator, MZ3 is by far the best antifouling membrane amongst the prepared membranes (Figure 12). These observations further demonstrate the influence of zwitterion inclusion in the performance of these membranes also extends to their fouling mitigation properties.

## 4. Conclusions

The successful synthesis of the zwitterion p(MAO-DMPA) via ring-opening and subsequent blending with PVDF membranes via phase inversion is presented. A series of membranes with various zwitterion contents were prepared, and the effect of this on membrane performance was investigated. The gel-like property of the zwitterion resulted in increasing membrane heterogeneity at higher loading as a result of increasing immiscibility with the polymer matrix, resulting in an unexpected trend reversal in membrane internal structure, water contact angle (WCA), and porosity. Overall, the inclusion of zwitterion enhanced flux, fouling, and dye rejection properties compared to the pristine PVDF. It was found that membranes with intermediate zwitterion content showed the best overall performance characteristics with respect to pure water flux and fouling propensity. Two solute rejection mechanisms were observed for CR based on internal membrane structure and zwitterion content for RG. The membrane that showed highest antifouling profile, highest flux, and acceptable solute rejection was MZ3, at 1.5 wt.% of zwitterion. The effects of zwitterion content on the evolution of internal membrane structure and how this affects rejection mechanisms was shown in this study, and the results bode well for future strategies in controlling membrane properties. 

## Figures and Tables

**Figure 1 membranes-10-00323-f001:**
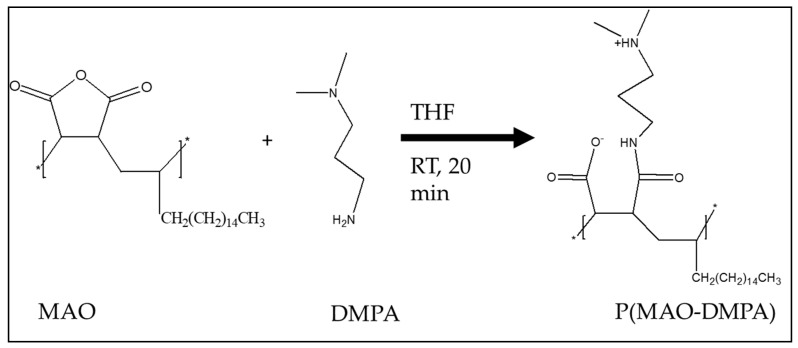
Synthesis of p(MAO-DMPA) by ring-opening reaction [23]. RT: room temperature.

**Figure 2 membranes-10-00323-f002:**
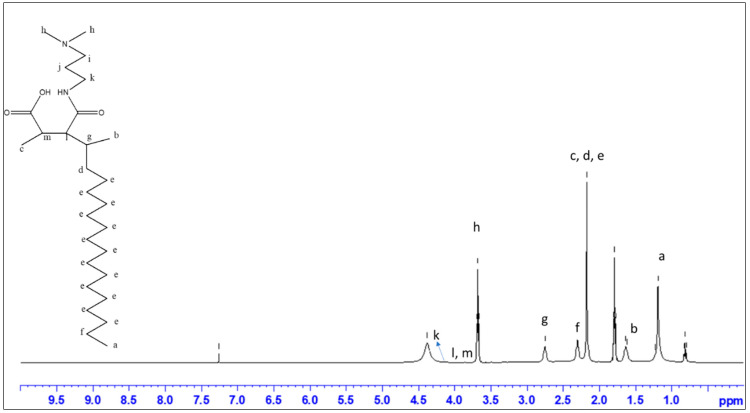
^1^H-NMR spectrum of the zwitterion p(MAO-DMPA).

**Figure 3 membranes-10-00323-f003:**
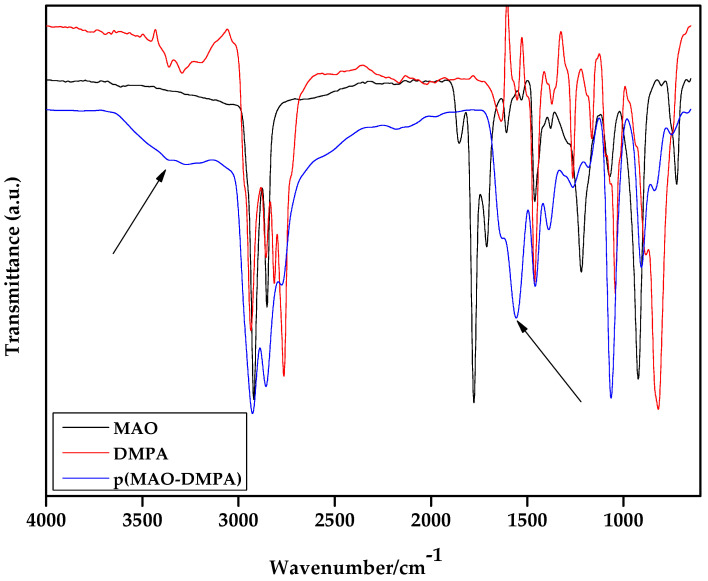
Attenuated total reflectance Fourier transform infrared spectroscopy (ATR-FTIR) spectra of the monomers MAO and DMPA together with that of the synthesized zwitterion, p(MAO-DMPA).

**Figure 4 membranes-10-00323-f004:**
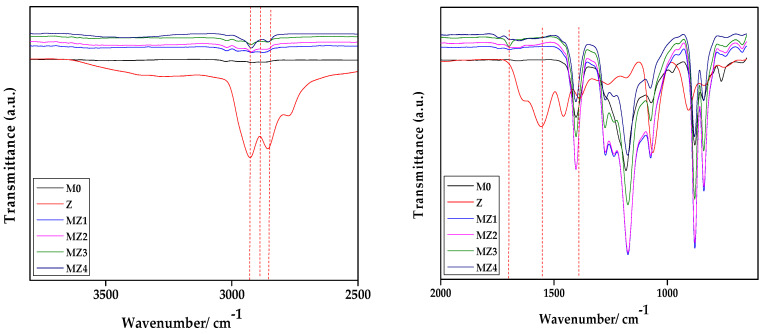
ATR-FTIR spectra of the PVDF and (MAO-DMPA)/PVDF composite membranes at different zwitterion contents (wt.%).

**Figure 5 membranes-10-00323-f005:**
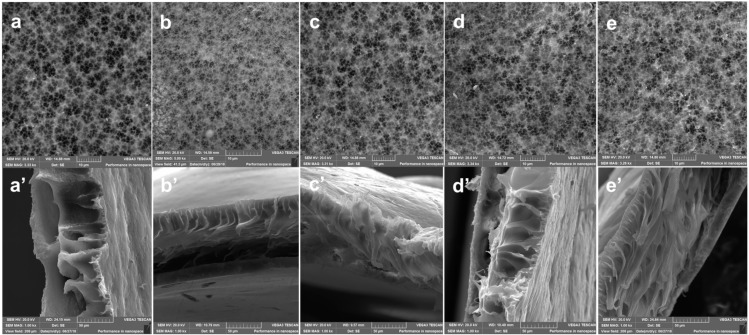
Surface and cross-section morphology of the prepared membranes.

**Figure 6 membranes-10-00323-f006:**
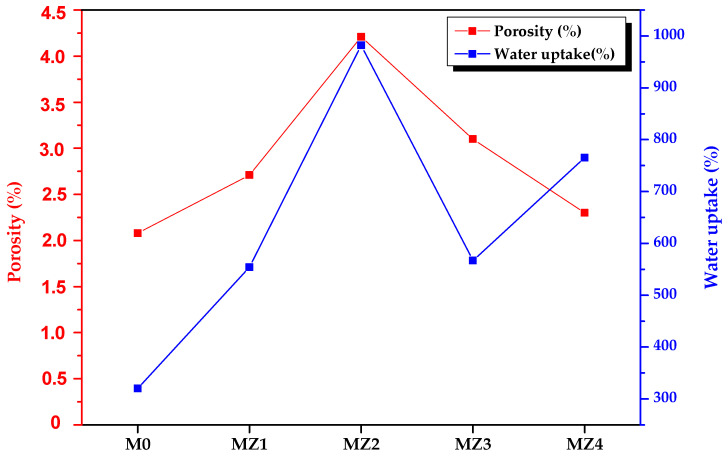
The measured water uptake capacity and calculated porosity of the prepared membranes.

**Figure 7 membranes-10-00323-f007:**
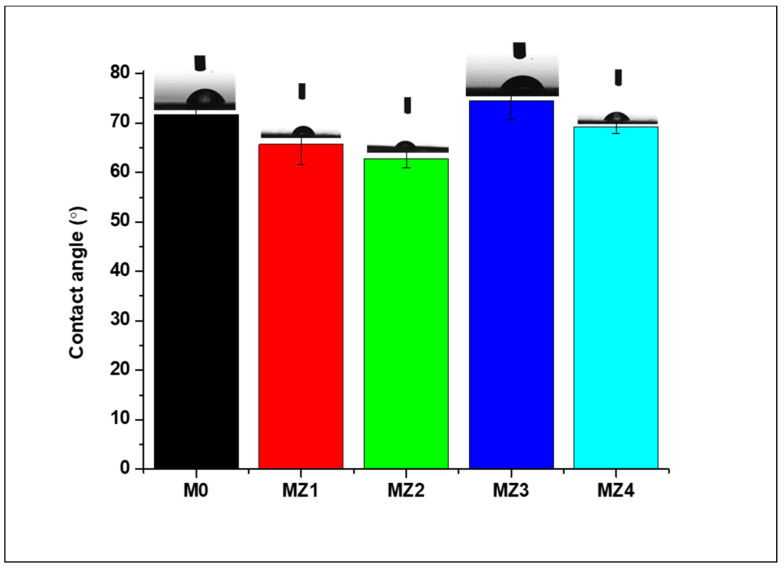
Variation of water contact angle with zwitterion content in the prepared membranes.

**Figure 8 membranes-10-00323-f008:**
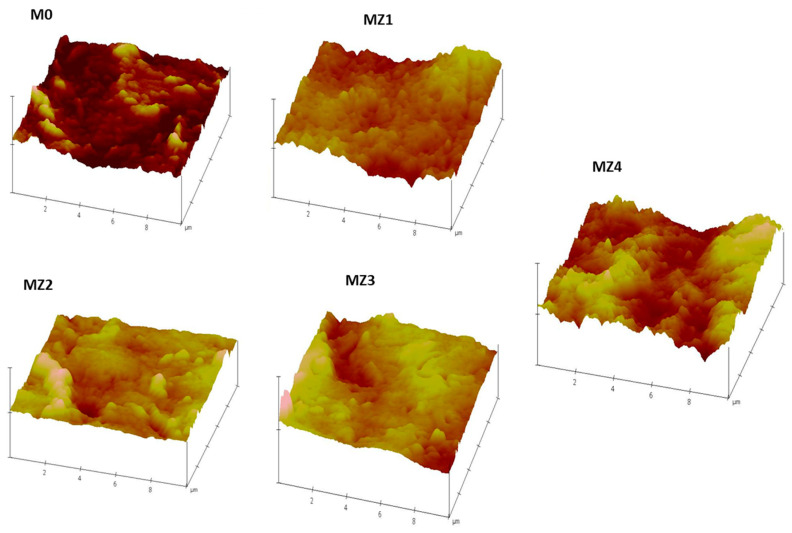
Atomic force microscopy (AFM) micrographs of the prepared membranes showing surface morphology variation with varying zwitterion content.

**Figure 9 membranes-10-00323-f009:**
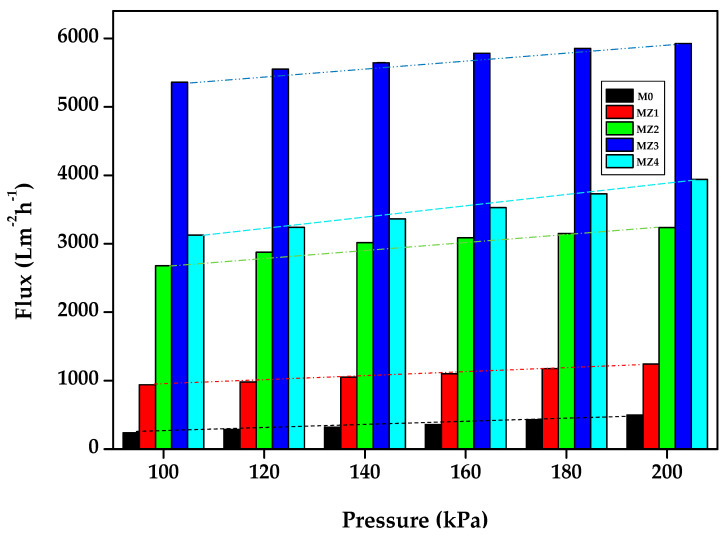
Pure water flux and permeability of the prepared membranes.

**Figure 10 membranes-10-00323-f010:**
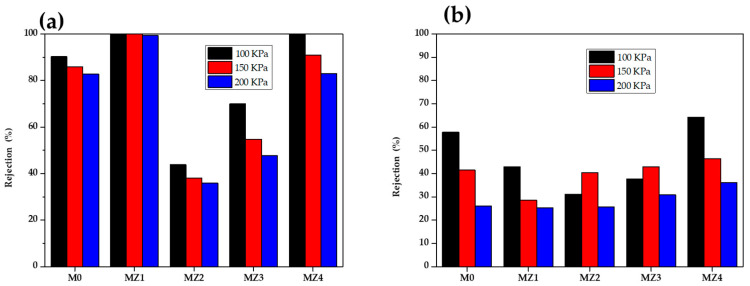
Rejection performance of the prepared membranes for (**a**) Congo red and (**b**) rhodamine 6G rejection at three different applied pressures.

**Figure 11 membranes-10-00323-f011:**
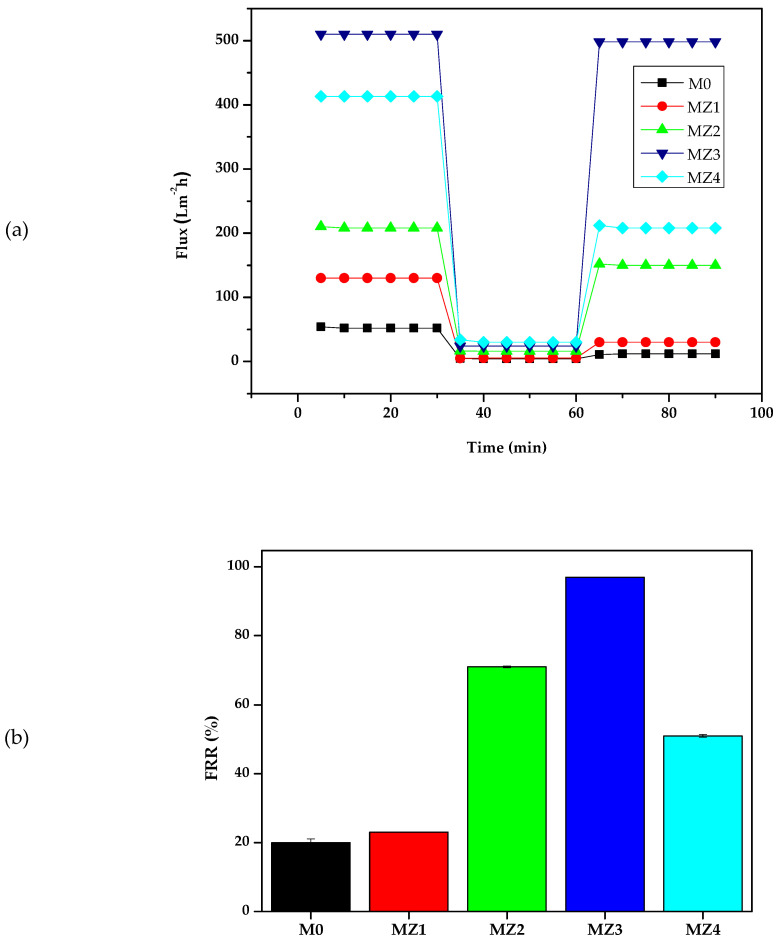
(**a**) Flux-fouling-recovery cycle; (**b**) the resultant water flux recovery ratio for the prepared membranes.

**Figure 12 membranes-10-00323-f012:**
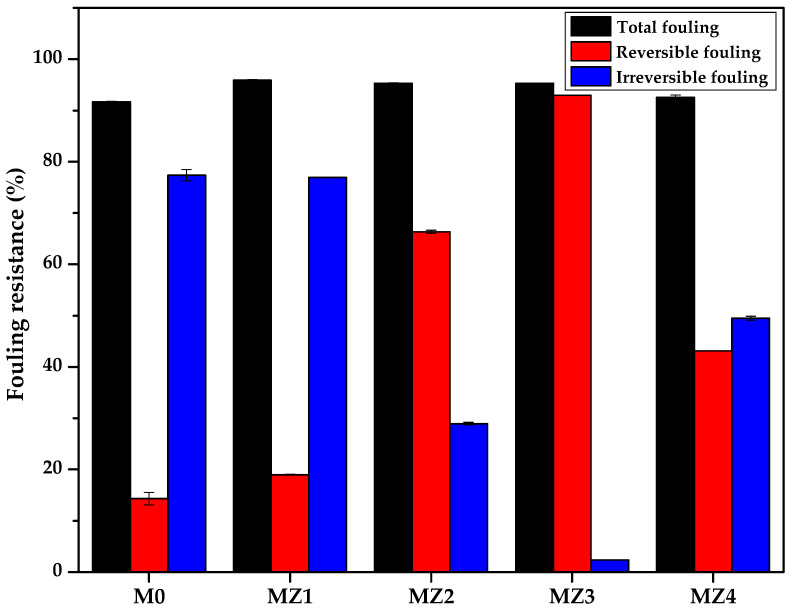
Fouling parameters of the fabricated membranes.

**Table 1 membranes-10-00323-t001:** Composition (wt.%) of the membrane casting solutions. PVDF: poly(vinylidene fluoride); PEG: poly(ethylene glycol); p(MAO-DMPA): poly-(maleic anhydride-alt-1-octadecene-3-(dimethylamino)-1-propylamine); NMP: *N*-methyl-2-pyrrolidine.

Description	Membrane ID	PVDF (wt.%)	PEG (wt.%)	P(MAO-DMPA) (wt.%)	NMP (wt.%)
M0	PVDF/PEG	16.00	2.00	------	82.00
MZ1	PVDF/PEG/Z_1_	16.00	2.00	0.50	81.50
MZ2	PVDF/PEG/Z_2_	16.00	2.00	1.00	81.00
MZ3	PVDF/PEG/Z_3_	16.00	2.00	1.50	80.50
MZ4	PVDF/PEG/Z_4_	16.00	2.00	2.00	80.00

**Table 2 membranes-10-00323-t002:** Calculated water uptake and porosity as well as measured membrane thickness and pore sizes.

Membranes	Water Uptake (%)	Porosity (%)	Pore Size (um)	Thickness (µm)
MZ0	320	2.08	0.342	92.12
MZ1	554	2.71	0.235	97.18
MZ2	982	4.21	0.169	94.40
MZ3	567	3.10	0.150	96.01
MZ4	765	2.30	0.156	129.11

**Table 3 membranes-10-00323-t003:** Surface roughness parameters for the prepared membranes.

Membranes	Roughness Parameters	
	R_a_ (nm)	R_q_ (nm)
M0	37.8	46.8
MZ1	60.7	77.4
MZ2	56.2	78.8
MZ3	54.1	71.7
MZ4	28.3	34.3

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
