# Peer review of "Effect of the Zwitterion, p(MAO-DMPA), on the Internal Structure, Fouling Characteristics, and Dye Rejection Mechanism of PVDF Membranes"

_membranes, 2020, doi:10.3390/membranes10110323_

Round 1

Reviewer 1 Report

The manuscript “Effect of the zwitterion, p(MAO-DMPA), on PVDF membranes internal structure, fouling characteristics and dye rejection mechanism” by Nelisa Gaxela et. al. is on a relevant topic of research. i.e. the application of membranes for dye separation by blending the zwitterions with PVDF powder. Several issues need to be discussed in this manuscript

  1. In this manuscript,Table 3 shows the pore size of the membrane is 0.1-0.3um, which should be classified into microfiltration membrane, not ultrafiltration membrane. So more evidence should be presented to prove that this membrane is belong to ultrafiltration.
  2. In Fig.10(b), the dye rejection of rhodamine is only 60% for the zwitterions modified membrane. As far as I am concerned, the modified membrane is not good enough to separate dyes because of the relatively low dye rejection rate.
  3. The research of dye separation is well understood, so where is the advantage of your research?
  4. In Figs 10-12. The modified membrane has a high dye rejection, but the antifouling performance and flux recovery ratio of which are relatively low. So the performance of the modified membrane needs to be further improved.

Author Response

Responses contained in the submitted document.

Reviewer 2 Report

The manuscript ‘Effect of the zwitterion, p(MAO-DMPA), on PVDF membranes internal structure, fouling characteristics and dye rejection mechanism’ refers to a very interesting issue involving the prepare of polymer membranes characterized by new properties, but I believe that manuscript in present form, should not be published in the Membranes.

It will be able to be published after considering the comments below:

The Results and discussion section should be corrected.

Line 35, Authors should add several references concerning membranes roughness. References below are strongly recommended.

Kowalik-Klimczak A., Gierycz P., Water Science & Technology 76/11 (2017) 3135-3141.

Boussu K., Vandecasteele B., Van der Bruggen B., Journal of Membrane Science 310 (2008) 51-65.

Boussu K., Belpaire A., Volodin A., Van Haesendonck C., Vand der Meeren P., Vandecasteele B., Van der Bruggen B., Journal of Membrane Science 289 (2007) 220-230.

Line 112, The work concern a flat-sheet polymer membranes. These membranes are used in spiral wound modules in pilot and industrial scale. The backwashing of spiral wound module damages the membrane. That is why, the discussion should be corrected.

Leaching is one of the most important factor to consider during preparation of membranes. The modifying agent leaching out from the membrane no longer functional or at least not as functional as it was before. Particularly, the modifying agent depending on its properties might be the reason of contamination of filtrate, which is undesirable effect. Please, explain what the stability of the new type of membrane developed in this work.

Author Response

(The authors gave the same response as above.)

Round 2

Reviewer 2 Report

Leaching is one of the most important factor to consider during preparation of membranes. That is why, I think an authors should be interested of this problem in futher work.